# A LEARNED REPRESENTATION FOR SCALABLE VECTOR GRAPHICS

**Raphael Gontijo Lopes,**[*] **David Ha, Douglas Eck, Jonathon Shlens**
Google Brain
{iraphael, hadavid, deck, shlens}@google.com

## ABSTRACT

Dramatic advances in generative models have resulted in near photographic quality for artificially rendered faces, animals and other objects in the natural world. In spite of such advances, a higher level understanding of vision and imagery does not arise from exhaustively modeling an object, but instead identifying higher-level attributes that best summarize the aspects of an object. In this work we attempt to model the drawing process of fonts by building sequential generative models of vector graphics. This model has the benefit of providing a scale-invariant representation for imagery whose latent representation may be systematically manipulated and exploited to perform style propagation. We demonstrate these results on a large dataset of fonts and highlight how such a model captures the statistical dependencies and richness of this dataset. We envision that our model can find use as a tool for designers to facilitate font design.

## 1 INTRODUCTION

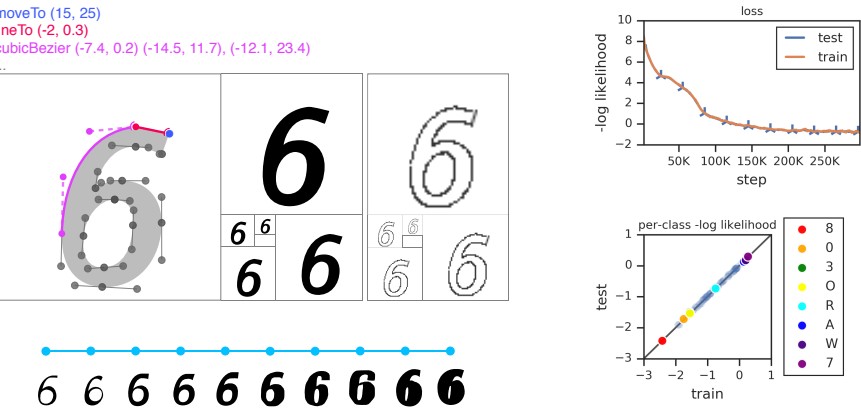

Figure 1: **Learning fonts in a native command space**. (left) Unlike pixels, scalable vector graphics (SVG) are scale-invariant representations whose parameterizations may be systematically adjusted to convey different styles. (right) We learn a latent representation of the SVG specification (Ferraiolo, 2001) that enables such manipulations. All images are samples from this generative model.

The last few years have witnessed dramatic advances in generative models of images that produce near photographic quality imagery of human faces, animals, and natural objects (Radford et al., 2015; Goodfellow et al., 2014; Brock et al., 2018; Karras et al., 2018; Kingma & Dhariwal, 2018). These models provide an exhaustive characterization of natural image statistics (Simoncelli & Olshausen, 2001) and represent a significant advance in this domain. However, these advances in image synthesis ignore an important facet of how humans interpret raw visual information (Reisberg & Snavely, 2010), namely that humans seem to exploit *structured* representations of visual

---

[*]Work done as a member of the Google AI Residency Program (g.co/airesidency)

concepts (Lake et al., 2017; Hofstadter, 1995). Structured representations may be readily employed to aid generalization and efficient learning by identifying higher level primitives for conveying visual information (Lake et al., 2015) or provide building blocks for creative exploration (Hofstadter, 1995; Hofstadter & McGraw, 1993). This may be best seen in human drawing, where techniques such as *gesture drawing* emphasize parsimony for capturing higher level semantics and actions with minimal graphical content (Stanchfield, 2007).

In this work, we focus on an subset of this domain where we think we can make progress and improve the generality of the approach. Font generation represents a 30 year old problem posited as a constrained but diverse domain for understanding high level perception and creativity (Hofstadter, 1995). Early research attempted to heuristically systematize the creation of fonts for expressing the identity of characters (e.g. `a`, `2`) as well as stylistic elements constituting the "spirit" of a font (Hofstadter & McGraw, 1993). Despite providing great inspiration, the results were limited by a reliance on heuristics and a lack of a learned, structured representation (Rehling, 2001). Subsequent work for learning font representations focused on models with simple parameterizations (Lau, 2009), template matching (Suveeranont & Igarashi, 2010), example-based hints (Zongker et al., 2000), or more recently, learning manifolds for geometric annotations (Campbell & Kautz, 2014).

We instead frame the problem of generating fonts by specifying it with Scalable Vector Graphics (SVG) – a common file format for fonts, human drawings, designs and illustrations (Ferraiolo, 2001). SVGs are a compact, scale-invariant representation that may be rendered on most web browsers. SVGs specify an illustration as a sequence of a higher-level commands paired with numerical arguments (Figure 1, top). We take inspiration from the literature on generative models of images in rasterized pixel space (Graves, 2013; Van den Oord et al., 2016). Such models provide powerful auto-regressive formulations for discrete, sequential data (Hochreiter & Schmidhuber, 1997; Graves, 2013; Van den Oord et al., 2016) and may be applied to rasterized renderings of drawings (Ha & Eck, 2017). We extend these approaches to the generation of sequences of SVG commands for the inference of individual font characters. The goal of this work is to build a tool to learn a representation for font characters and style that may be extended to other artistic domains (Clouâtre & Demers, 2019; Sangkloy et al., 2016; Ha & Eck, 2017), or exploited as an intelligent assistant for font creation (Carter & Nielsen, 2017).

Our main contributions are: 1) Build a generative model for scalable vector graphics (SVG) images and apply this to a large-scale dataset of 14 M font characters. 2) Demonstrate that the generative model provides a latent representation of font styles that captures a large amount of diversity and is consistent across individual characters. 3) Exploit the latent representation from the model to infer complete SVG fontsets from a single character. 4) Identify semantically meaningful directions in the latent representation to globally manipulate font style.

## 2 METHODS

We compiled a font dataset composed of 14 M examples across 62 characters (i.e. `0-9`, `a-z`, `A-Z`), which we term `SVG-Fonts`. The dataset consists of fonts in a common font format (SFD)[1] converted to SVG, excluding examples where the unicode ID does not match the targeted 62 character set specified above. In spite of the filtering, label noise exists across the roughly 220 K fonts examined.

The proposed model consists of a variational autoencoder (VAE) (Kingma & Welling, 2013; Ha & Eck, 2017) and an autoregressive SVG decoder implemented in Tensor2Tensor (Vaswani et al., 2018). Briefly, the VAE is a convolutional encoder and decoder paired with instance normalization conditioned on the label (e.g. `a`, `2`, etc.) (Dumoulin et al., 2017; Perez et al., 2018). The VAE is trained as an class-conditioned autoencoder resulting in a latent code $z$ that is largely class-independent (Kingma et al., 2014). The latent $z$ is composed of $\mu$ and $\sigma$: the mean and standard deviation of a multivariate Gaussian. The SVG decoder consists of 4 stacked LSTMs (Hochreiter & Schmidhuber, 1997) trained with dropout (Srivastava et al., 2014; Zaremba et al., 2014; Semeniuta et al., 2016) and a Mixture Density Network (MDN) at its final layer. The LSTM receives as input the previous sampled MDN output, concatenated with the discrete class label and the latent style

---

[1] https://fontforge.github.io

Figure 2: **Exploiting the latent representation of style**. (left) Examples generated by sampling a random latent representation $z$ and running the SVG decoder by conditioning on $z$ and all class labels. The learned $z$ is class-agnostic and covers a wide range of font styles. (right) Examples generated by computing $z$ from a *single* character (purple box) and generating SVG images for all other characters in a font. A single character may provide sufficient information for reconstructing the rest of a font set. Each character is selected as the best of 10 samples.

representation $z$. The SVG decoder's loss is composed of a softmax cross-entropy loss between over one-hot SVG commands plus the MDN loss applied to the real-valued arguments.

In principle, the model may be trained end-to-end, but we found it simpler to train the two parts of the model separately. Note that both the VAE and MDN are probabilistic models that maybe sampled many times during evaluation. The results shown here are the selected best out of 10 samples.

# 3 RESULTS

We compiled the `SVG-Fonts` dataset wherein individual SFD font characters were normalized and converted into SVG format for training and evaluation. We trained a VAE and SVG decoder over 3 epochs of the data and evaluated the results on a hold-out test split. Over the course of training, we find that the model does indeed improve in terms of likelihood and plateaus in performance, while not overfitting on the training set (Figure 1, top left). Yet, we note a small but systematic spread in average likelihood across classes (Figure 1, bottom left). What follows is an analysis of the representational ability of the model to learn and generate SVG specified fonts.

## 3.1 EXPLOITING THE LATENT REPRESENTATION FOR STYLE PROPAGATION

We first ask whether the proposed model may learn a latent representation of font style that captures a large amount of diversity. We demonstrate this by generating SVG characters using the SVG decoder, while conditioning on a randomly sampled $z$. In Figure 2 (left) we see that the decodings represent a wide array of font styles.

Because the VAE is conditioned on the class label, we expect that the latent representation $z$ would only encode the font style with minimal class information (Kingma et al., 2014). We wish to exploit this model structure to perform style propagation across fonts. In particular, we ask whether a *single character* from a font set is sufficient to infer the rest of the font set in a visually plausible manner (Rehling, 2001; Hofstadter & McGraw, 1993).

To perform this task, we calculate the latent representation $z$ for a single character and condition the SVG decoder on $z$ as well as the label for all other font characters (i.e. `0-9`, `a-z`, `A-Z`). Figure 2 (right) shows the results of this experiment. For each row, $z$ is calculated from the character in the red box. The other characters in that row are generated from the SVG decoder conditioned on $z$.

We observe a perceptually-similar style consistently within each row. Note that there was no requirement during training that the same point in latent space would correspond to a perceptually similar character across labels – that is, the consistency across class labels was learned in an unsu-

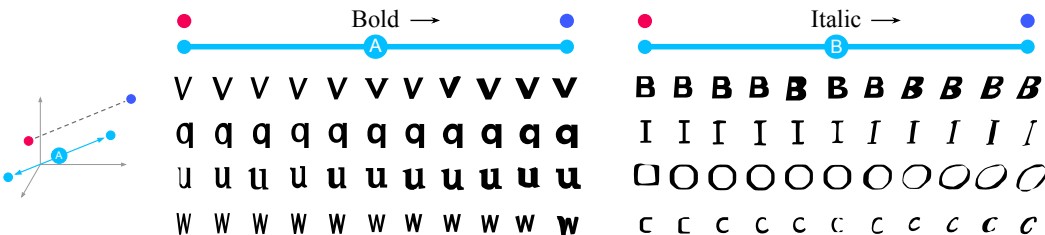

Figure 3: **Building style analogies with the learned representation**. Semantically meaningful directions may be identified for globally altering font attributes. For instance, bold (blue) and non-bold (red) examples of a concept provide a vector direction in the latent space (left) that may be added to arbitrary points (yellow) for decreasing or increasing the strength of the attribute.

pervised manner (Rehling, 2001; Hofstadter & McGraw, 1993). Thus, a single value of $z$ seems to correspond to a perceptually-similar set of characters that resembles a plausible fontset.

Additionally, we observe a large amount of style variety across rows (i.e. different $z$) in Figure 2 (right). The variety indicates that the latent space $z$ is able to learn and capture a large diversity of styles observed in the training set. Finally, we also note that for a given column the decoded glyph does indeed belong to the class that was supplied to the SVG decoder. These results indicate that $z$ encodes style information consistently across different character labels, and that the proposed model largely disentangles class label from style.

### 3.2 BUILDING STYLE ANALOGIES WITH THE LEARNED REPRESENTATION

Given that the latent style is perceptually smooth and aligned across class labels, we next ask if we may find semantically meaningful directions in this latent space. In particular, we ask whether these semantically meaningful directions may permit global manipulations of font style.

Inspired by the work on word vectors (Mikolov et al., 2013), we ask whether one may identify analogies for organizing the space of font styles (Figure 3, top). To address this question, we select positive and negative examples for semantic concepts of organizing fonts (e.g. bold) and identify regions in latent space corresponding to the presence or absence of this concept (blue and red points). We compute the average $z_{red}$ and $z_{blue}$, and define the concept direction $c = z_{blue} - z_{red}$.

We test if these directions are meaningful by taking an example font style $z^*$ from the dataset (Figure 3, right, yellow), and adding (or subtracting) the concept vector $c$ scaled by some parameter $\alpha$. Finally, we compute the SVG decodings for $z^* + \alpha c$ across a range of $\alpha$.

Figure 3 shows the resulting fonts. Note that across the three properties examined, we observe a smooth interpolation in the direction of the concept modeled (e.g.: first row v becomes increasingly bold from left to right). We take these results to indicate that one may interpret semantically meaningful directions in the latent space. Additionally, these results indicate that one may find directions in the latent space to globally manipulate font style.

## 4 DISCUSSION

In the work we presented a generative model for vector graphics. This model has the benefit of providing a scale-invariant representation for imagery whose latent representation may be systematically manipulated and exploited to perform style propagation. We demonstrate these results on a large dataset of fonts and highlight the limitations of a sequential, stochastic model for capturing the statistical dependencies and richness of this dataset. Even in its present form, the current model may be employed as an assistive agent for helping humans design fonts in a more time-efficient manner (Carter & Nielsen, 2017; Rehling, 2001).

