# OpenReview forum: "A Learned Representation for Scalable Vector Graphics"
_ICLR.cc/2019/Workshop/DeepGenStruct — DeepGenStruct 2019_

### Official Review · AnonReviewer1 · 2019-04-09
**Good idea for generating character images**

**Rating:** 3
**Confidence:** 2

**Review:**

This paper provided an interesting idea for generating character (letters and digits) images of various fonts. The proposed model consists of a VAE (as a class-conditioned encoder)  and a SVG decoder (with LSTM and MDN layers) and can generate reasonable images for digits and letters.

Pros:

1. The generated characters are good. We can easily recognize the generated characters. And they are of various fonts.

2. Since the encoder is class-conditioned, we can generate other characters of the same font given one character.

3. The learned latent embeddings are meaningful.

Cons & questions:

1. The description of the model is not very clear. It is better to add more details.

2. Where are the references?

---

### Official Review · AnonReviewer2 · 2019-04-12
**Good application of generative models, well written**

**Rating:** 4
**Confidence:** 2

**Review:**

This paper proposes deep generative model consisting of VAE + autoregressive LSTM decoder for learning to generate fonts in SVG (scalable vector graphics format)

The paper is well written, well motivated. Perhaps it would be great to see some plot of different hyperparameters and different modeling choices and its effect on results. Additionally, it would be great if authors could elaborate on details of cost function that they used (I would imagine that it is a mix of classifier that predicts actions like moveTo/lineTo etc and well as regression module that predicts coordinates).

Overall it looks like a well executed paper.
I recommend accept

---

### Decision · Program_Chairs · 2019-04-19
**Acceptance Decision**

**Decision:**

Accept

**Comment:**

Accepted